# Transcriptional and Ultrastructural Analyses Suggest Novel Insights into Epithelial Barrier Impairment in Celiac Disease

**DOI:** 10.3390/cells9020516

**Published:** 2020-02-24

**Authors:** Agnieszka Sowińska, Yasser Morsy, Elżbieta Czarnowska, Beata Oralewska, Ewa Konopka, Marek Woynarowski, Sylwia Szymańska, Maria Ejmont, Michael Scharl, Joanna B. Bierła, Marcin Wawrzyniak, Bożena Cukrowska

**Affiliations:** 1Department of Pathology, The Children Memorial Health Institute, Al. Dzieci Polskich 20, 04-730 Warsaw, Poland; a.sowinska@ipczd.pl (A.S.); e.czarnowska@ipczd.pl (E.C.); e.konopka@ipczd.pl (E.K.); s.szymanska@ipczd.pl (S.S.); m.ejmont@ipczd.pl (M.E.); b.cukrowska@ipczd.pl (B.C.); 2Department of Gastroenterology and Hepatology, University Hospital Zürich, CH-8001 Zürich, Switzerland; michael.scharl@uzh.ch (M.S.); marcin.wawrzyniak@uzh.ch (M.W.); yasser.morsy@uzh.ch (Y.M.); 3Clinic of Gastroenterology, The Children Memorial Health Institute, Al. Dzieci Polskich 20, 04-730 Warsaw, Poland; b.oralewska@ipczd.pl (B.O.); marek.woynarowski@czmdm.pl (M.W.)

**Keywords:** celiac disease, epithelium, epithelial barrier, tight junctions, adherence junctions, fetal like type enterocytes

## Abstract

Disruption of epithelial junctional complex (EJC), especially tight junctions (TJ), resulting in increased intestinal permeability, is supposed to activate the enhanced immune response to gluten and to induce the development of celiac disease (CD). This study is aimed to present the role of EJC in CD pathogenesis. To analyze differentially expressed genes the next-generation mRNA sequencing data from CD326+ epithelial cells isolated from non-celiac and celiac patients were involved. Ultrastructural studies with morphometry of EJC were done in potential CD, newly recognized active CD, and non-celiac controls. The transcriptional analysis suggested disturbances of epithelium and the most significant gene ontology enriched terms in epithelial cells from CD patients related to the plasma membrane, extracellular exome, extracellular region, and extracellular space. Ultrastructural analyses showed significantly tighter TJ, anomalies in desmosomes, dilatations of intercellular space, and shorter microvilli in potential and active CD compared to controls. Enterocytes of fetal-like type and significantly wider adherence junctions were observed only in active CD. In conclusion, the results do not support the hypothesis that an increased passage of gluten peptides by unsealing TJ precedes CD development. However, increased intestinal permeability due to abnormality of epithelium might play a role in CD onset.

## 1. Introduction

Celiac disease (CD) is an autoimmune-mediated disorder with characteristic histological changes in the small intestine and serum specific antibodies, which is triggered by the ingestion of gluten from wheat, barley, and rye in genetically predisposed individuals [1]. CD development requires the presence of gluten intestinal tissue transglutaminase 2 (tTG2)—the enzyme modifying gluten peptides, and the genes encoding human leukocyte antigen (HLA)-DQ2 or -DQ8 [2]. The gluten from food products after digestion in the intestinal lumen generates small peptides, which then are transferred through the epithelial barrier into lamina propria. This passage can take place using two different routes: the transcellular transport mediated by endocytosis on the luminal side of enterocytes or the paracellular transfer depend on the epithelial junctional complex (EJC). The binding of deamidated gluten proteins (DPG) to antigen-presenting cells amplifies gluten-specific T cell response that finally results in increased amounts of intraepithelial lymphocytes, crypt hyperplasia, and villous atrophy, as well as B cell stimulation and the production of antibodies directed against DGP and tTG2 [3]. Thus, mechanisms of CD development are well known, and gluten is the main external trigger for CD autoimmunity. However, its intake does not fully explain CD pathogenesis. Although gluten is introduced into the diet in early childhood, the CD has ceased to be a disease of young children and can develop at any age, including older people. That is why there is a hypothesis that early disruption of the gut epithelial barrier could precede the onset of gluten-induced immune events [4]. It is supposed that the dysfunction of tight junctions (TJ)—the most apical unit between neighboring epithelial cells—induces the transport of immunogenic gluten peptides into lamina propria and increases activation of the immune response leading to CD development [5]. Fasano et al. showed that zonulin—a prehaptoglobulin 2—is the primary regulator of gut permeability [6], and its upregulation is dependent on both intestinal bacteria [7] and gliadin, a prolamine component of gluten [8,9]. Zonulin signaling leads to disassembling of junctional proteins and cytoskeleton rearrangement resulting in the disruption of TJ [6].

There are several studies mainly based on immunohistochemical analyses of intestinal specimens demonstrating over-expression of the pore-forming protein claudin-2 and downregulation of pore–sealing proteins claudin-3 and 4, as well as scaffold protein zonula occludens (ZO)-1 in CD patients [10,11]. The recent transmission electron microscopy analyses on duodenal biopsies of CD patients have shown dilatation and destruction of TJ structures [12,13]. However, there is a lack of comprehensive studies devoted to the ultrastructural morphology of intestinal epithelium and EJC including not only TJ but also adherence junctions (AJ) and desmosomes in CD patients.

Our study aims to present transcriptional mRNA next-generation sequencing (mRNA NGS) analysis of epithelial cells and ultrastructural research of enterocytes and EJC, and search whether abnormalities of gut epithelium play a role in CD pathogenesis.

## 2. Materials and Methods

### 2.1. Filtering Criteria of Public RNA-Seq Datasets 

An NCBI BioProject and Sequence Read Archive (SRA) was used to find relevant public available RNA-seq projects related to CD. The list of the projects is presented in Table 1. As RNA-seq data analysis highly dependent on the purity of the samples, and the majority of samples are biopsies from the small intestine, we selected only one data set with the accession number “PRJNA327491,” where epithelial cells within the biopsies were enriched using CD326 as a marker.

### 2.2. RNA-Seq Analysis

An SRA tool kit (version 2.8.2) was used to download fastq files and FastQC (version 0.11.5) to check the data quality. The human genome GRCh38 from Ensemble was used as a reference genome for STAR (version 2.5.4) for mapping the reads. The EdgeR package (version 3.26.8) [14,15] was used for detecting differentially expressed genes. A principal component analysis was performed across all the samples using the RPKM count. A heatmap was generated from unsupervised clustering using differentially expressed genes. Gene ontology (GO) was carried out using Database for Annotation, Visualization, and Integrated Discovery (DAVID) tool version 6.8. [16,17].

### 2.3. Patients’ Biopsy Samples

Biopsies were taken during endoscopy and histological analyses were performed by two independent pathologists using the modified Marsh–Oberhuber classification [18]. The active CD was recognized in accordance with the current guidelines of European Society for Pediatric Gastroenterology Hepatology and Nutrition (ESPGHAN) from 2012, i.e., in patients with positive immunoglobulin (Ig) A antibodies against tTG2 (anti-tTG2-IgA) or in case of IgA deficit with positive anti-tTG2-IgG (all tests from ThermoScientific, Phadia AB, Uppsala, Sweden), and with histological changes described at least as Marsh 2 [19]. The potential CD was diagnosed in patients with positive anti-tTG2-IgA and normal histological picture of small intestine specimens (Marsh 0) or histological changes described as Marsh 1 (an increase in the number of IEL > 25/100 enterocytes). In addition, antibodies against endomysium (anti-EMA-IgA) with the use of indirect immunofluorescence assay (Euroimmun, Lubeck, Germany) were determined in all patients with potential CD. A titer of ≥1:5 was considered as a positive result.

### 2.4. Ultrastructural Analyses

Ultrastructural studies were done in 12 pediatric patients, and this group included 3 patients with potential CD. The characteristics of patients are presented in Table 2. The control group consisted of 9 children (6 boys and 3 girls) age-matched to CD group with normal histology (Marsh 0) and the lack of serum CD antibodies in whom finally functional gut disorders were diagnosed.

The tissues obtained during endoscopy were fixed in 4% glutaraldehyde in phosphate buffer solution and postfixed in 1% osmium tetraoxide (Sigma-Aldrich, Saint Louis, MO, USA), then successively dehydrated in ascendant ethanol and propylene oxide (EMS, Hatfield, PA, USA), and finally were embedded in Epon resin (EMS, Hatfield, PA, USA). According to semi-thin sections the longitudinally dissected villi were trimmed and ultrathin cut sections were stained with uranyl acetate and lead citrate (EMS, Hatfield, PA, USA) and examined by transmission electron microscopy (Jem 1011; Jeol, Peabody, MA, USA). The ultrastructural features of the intestine structures and junctions were visualized at magnifications ranging from ×5000 to ×60,000. The widths and lengths of intercellular junctions and microvilli were measured using the iTEM morphometric program (Olympus, Munich, Germany) at a magnification of × 60,000. Ultrastructural analyses were performed in 10 selected tissue areas, and only epithelial structures that were precisely longitudinally sectioned were measured. At least 5 values of each type of intercellular junction/patient and at least 3 values associated with microvilli/patient were obtained. The results were presented in nm.

### 2.5. Statistical Analysis

A generalized-linear model based on the negative binomial distribution was applied to the RNA-seq data analysis to estimate the dispersion parameter across all genes with the assumption of common dispersion for all genes. The default trimmed mean of M-values (TMM) method was used for normalization and adjustment of multiple comparisons (with a false discovery cut-off <0.05) was performed using the Benjamini–Hochberg correction.

Statistical analysis of ultrastructural measurements was performed with GraphPad Prism 8 software. Multiple measurements for each patient were used for a one-way ANOVA test with Tukey corrections for multiple comparisons. The maximum significance level for all analyses was set to α = 0.05.

### 2.6. Ethical Statement

The study was approved by a local Ethics Committee (No 227/KBE/2015), and written informed consent was obtained from the patients’ parents, caretakers, or patients aged ≥ 16 years old, concerning the use of their biopsy samples for scientific purposes. The study protocol conforms to the ethical guidelines of the 1975 Declaration of Helsinki.

## 3. Results

### 3.1. Transcriptional Analyses

CD326+ epithelial cells from active CD biopsies were compared with CD326+ epithelial cells from five biopsies of control patients. A total number of 1194 differentially expressed genes revealed a *p*-value < 0.05. Among 1194 differentially expressed genes, 882 genes were upregulated and 312 genes were downregulated in epithelial cells from CD patients as compared to controls. Out of those 1194 genes, a selection of 471 genes with a cut off fold change value of 2 or −2, are shown in Figure 1a. A principal component analysis (PCA) was carried out on the total number of genes using the count values generated from the count feature package.

PCA showed 33% of the variance between groups in the first component and 30% variance within the groups indicating the second component. A heatmap on the differentially expressed genes showed the same clustering of PCA (Figure 1b,c).

In the biological process category (Figure 2a), many genes were associated with GO terms response to immune response, inflammatory response, and regulation of the immune response. In the molecular function category (Figure 2b), genes associated with the terms antigen binding, endopeptidase activity, and MHC class II receptor activity were enriched. On the level of cellular compartments (Figure 2c), the most significant GO enriched terms were plasma membrane, extracellular exome, extracellular region, and extracellular space.

Results obtained with mRNA NGS analysis suggested disturbance of intestinal barrier on the level of epithelial cells. As epithelium and EJC can be easily visualized by transmission electron microscopy, further research was directed into ultrastructural analyses.

### 3.2. Ultrastructure of Enterocytes

Enterocytes of the control non-CD group had normal brush border, and the filamentous network of the terminal web with well-developed anchoring filaments (rootless) penetrated into the considerable depth (Figure 3a). In contrast, enterocytes of potential CD as well as active CD groups exhibited irregularly distributed and statistically significantly shorter microvilli (Figure 3b) with the heterogeneously organized anchoring filaments in the terminal web (Figure 3a). The microvilli in CD were covered by various thickness glycocalyx layer without significant differences compared to the control group. However, its discontinuity was visible. Intercellular spaces were increased with evident dilatation in both CD groups (Figure 3a), and protrusions of neighboring cells entered these spaces. Numerous enterocytes in active CD (histologically determined as Marsh 3) presented structures like canicular system located at the apical part of cells (called apical canicular system) characteristic for enterocytes of fetal-type (Figure 3a). They also showed an abundance of endosomes. Such enterocytes were not observed in the early phase of CD (Marsh 0-1) and the control group. Other structures, such as endoplasmic rough reticulum and polyribosomes were similar in morphology and number in all studied groups.

### 3.3. Enterocyte Intercellular Junctions

The apical EJC demonstrated significant tighter TJ both in potential and active CD compared to controls (Figure 4a,b). The average values (AV) of TJ widths were 9.06 nm, 9.11 nm, and 10.28 nm respectively for patients with potential CD, active CD, and non-CD controls. In contrast, AJ presented wider assembling, but only in active CD (AV = 26.24 nm) in comparison with controls (AV = 23.54 nm). The desmosomes showed significant irregular symmetry of morphology in both types of CD (Figure 4a), and their gap width was significantly wider in potential CD (AV = 33.38 nm) compared with controls (AV = 29.82 nm) (Figure 4b). There were no statistical differences in the lengths of TJ, AJ, and desmosomes between the studied groups (Figure 4c).

## 4. Discussion

As disruption of the small intestine epithelial barrier could be involved in CD development, we decided to investigate the epithelial cells from intestinal biopsies of CD patients on both transcriptional and ultrastructural levels. According to our knowledge, such comprehensive analyses of specific cell compartments have not been performed until now.

Our mRNA-seq data analysis of publicly available data of epithelial cells from CD in relation to non-CD controls show that epithelial CD326+ cells isolated from biopsies of CD patients differ in expression of 1194 genes, among which, 882 genes were upregulated and 312 genes were downregulated. Moreover, based on gene expression, these cells grouped together and were separated from non-CD control epithelial cells. Grouping of samples based on disease status was confirmed with hierarchical clustering of genes, and epithelial cells from non-CD controls and CD patients clustered separately. Then, the functional enrichment analysis of genes differentially expressed in small intestine epithelial cells was performed in three main categories that included genes associated with biological processes, cellular components, and molecular functions. The enrichment of genes involved in immune responses, antigen processing, and presentation in cells isolated from celiac patients has confirmed that CD is an autoimmune disease that occurs in MHC II genetically predisposed individuals. Interestingly, in the third tested category GO terms that describe plasma membrane, extracellular exosome, cytosol, membrane, extracellular region, extracellular space, integral component of plasma membrane, and apical plasma membrane were significantly enriched with most of the upregulated genes suggesting that epithelial cells could be involved in CD pathogenesis, and our ultrastructural research indeed demonstrated that epithelium exhibited significant differences between CD and non-CD patients.

Ultrastructural analysis of epithelial cells showed as previously presented by other authors [20] shorter microvilli with the disturbed organization of filaments anchoring the terminal web and abnormalities in the glycocalyx structure in CD. However, for the first time, we have described increased intercellular spaces with visible dilatation in potential and active CD as well as the presence of apical canicular systems similar to this present in fetal-like type enterocytes in active CD. Such a canicular system arises by invaginations of cell membranes in the base of microvilli forming vesicles of different sizes and tubules. It plays a role in the transport of antibodies and other intact proteins in the first hours of life [21]. These proteins are further carried out to the basolateral cell membrane and are released into intercellular space, and finally, reach the lymphatic vessels [22]. The presence of enterocytes, typical for the fetal period in active CD with villous atrophy, could suggest their intensive exchange or disturbed maturation. It is not clear whether the presence of the tubular system has an association with a significant expansion of the intercellular space between these cells, but enlarged intercellular spaces may be related to our transcriptional results showing an increase in the expression of genes responding to the extracellular matrix. In addition, these spaces between enterocytes in the early stage of the disease may have an effect on the impairment of the epithelial barrier, and as a consequence on the transport of gluten peptides into enlarged intercellular spaces and then into lamina propria.

Interestingly, we have also observed numerous endosomes in enterocytes of patients with villous atrophy. Experimental studies of epithelial transport pathways have shown that e.g., silver ions or horse reddish peroxidase at first occur in high-density endosomes, and later in TJ. Kersting et al. have suggested that these endosomes may lead to cell stress activating apoptotic death [23,24]. Furthermore, investigations have presented that antigens first are transferred by the transcellular route in endosomes, while later, after activation of mast cells, the paracellular pathway is involved in what finally results in the increased uptake of antigens [23,25].

Moreover, our ultrastructural analyses have shown that TJ, which controls the paracellular passage of ions, solutions, antigens, are significantly tighter in CD patients than in non-CD controls. Contrary to this result, Goswami et al. presented TJ dilatation and degenerations in TJ penta-laminar structures [12]. Interestingly, similar ultrastructural abnormalities with TJ dilatations were found in asymptomatic and serologically negative first-degree relatives of CD patients suggesting that epithelial barrier disruption could lead ahead to CD autoimmunity development [13]. Indeed, first degree relatives are the at-risk group for CD, but it should be emphasized that the incidence of CD in this group has not exceeded 11% [26]. Thus, it is not entirely clear whether the relatives of whom TJ measurements were performed developed CD. In addition, by analyzing representative ultrastructural images of CD relatives presented by Mishra et al., one can see that for measurements of TJ width cross-sections with lateral views were chosen, and such specimens should not be selected for morphometric analyses [13]. We would like to emphasize that in our study measurements of intercellular junctions were performed only on the precise longitudinal sections. The use of such techniques allowed us to find narrower TJ gaps both in active CD and in the very early phases of CD, i.e., in potential CD. Potential CD is characterized by the presence of CD specific autoantibodies in sera (patients enrolled to our study had positive celiac specific autoantibodies both anti-tTG2-IgA and anti-EMA-IgA) and normal histology of the small intestine (described as Marsh 0 in modified Marsh–Oberhuber classification) or only increased number of IEL without other features of inflammation (described as Marsh 1). Recently, a long-term study of 280 children with potential CD on a gluten-containing diet has shown that the cumulative incidence of progression to active CD with villous atrophy was only 43% [27]. However, in our study, 2 of 3 children were followed for 3 years, and one of them (patient No 1) developed active CD, while the other (patients No 2) had persistently elevated levels of anti-tTG2-IgA. Thus, potential CD diagnosed in children enrolled in our study seems to be an early phase of active CD.

Although no leakage of TJ in CD patients has been found, the ultrastructural results from other interepithelial junctions and increased intercellular spaces indicate damage to the epithelial barrier in the early phase of CD (potential CD) and active CD. Wider assembling of AJ was observed in active CD, and abnormalities of desmosomal structures were present in both potential and active CD. Some of these findings, especially those present only in active CD, may be secondary to inflammation process in the small intestine, but changes found in potential CD (without histological features of inflammation) could be responsible for increased permeability of epithelial barriers and gluten peptide transfer leading to active CD development. The question remains, what leads to disruption of the desmosome structure and extension of extracellular spaces in the epithelium in the early phase of CD. It is supposed that the disturbances of gut microbiota could play a role in this process [4]. An intestinal barrier should be recognized not only as an epithelial barrier but as a functional unit composed of three main components: microbial barrier, epithelial barrier, and immune barrier of gut-associated lymphoid tissue [4]. Thus, dysbiosis induced by infectious agents or antibiotics might be involved in an impairment of epithelium. Our earlier experimental studies on germ-free (GF) mice have shown that microbiota plays a significant role in the maturation and formation of EJC in the intestinal epithelium [28]. Interestingly, enterocytes of GF mice revealed abnormal desmosomes or their lack and increased width of AJ, so similar changes to those currently found in potential CD.

In summary, the results of the study show specific epithelium changes occurring early in CD, such as widened spaces between enterocytes and desmosome abnormalities, which can lead to unsealing the epithelial barrier. However, the results do not confirm the leakage of TJ. On the contrary, smaller gaps in TJ areas in CD patients than in controls may indicate tightness of epithelial barrier in the most apical part despite changes in the cell cytoskeleton and other interepithelial junctions to protect against the uptake of luminal antigens and to prevent back diffusion of substances from dilated intercellular space.

## 5. Conclusions

Our findings do not support the generally accepted hypothesis that increased paracellular trafficking of gluten peptides through disrupted TJ precedes CD development. Thus, it seems that TJ does not play such a significant role in an increased epithelial permeability in CD development. However, a leaky gut barrier due to abnormality of epithelium found on both transcriptional and ultrastructural levels might induce transport of gluten peptide into enlarged intercellular spaces and lamina propria, and enhance immune response to gluten in the early phase of CD.

## Figures and Tables

**Figure 1 cells-09-00516-f001:**
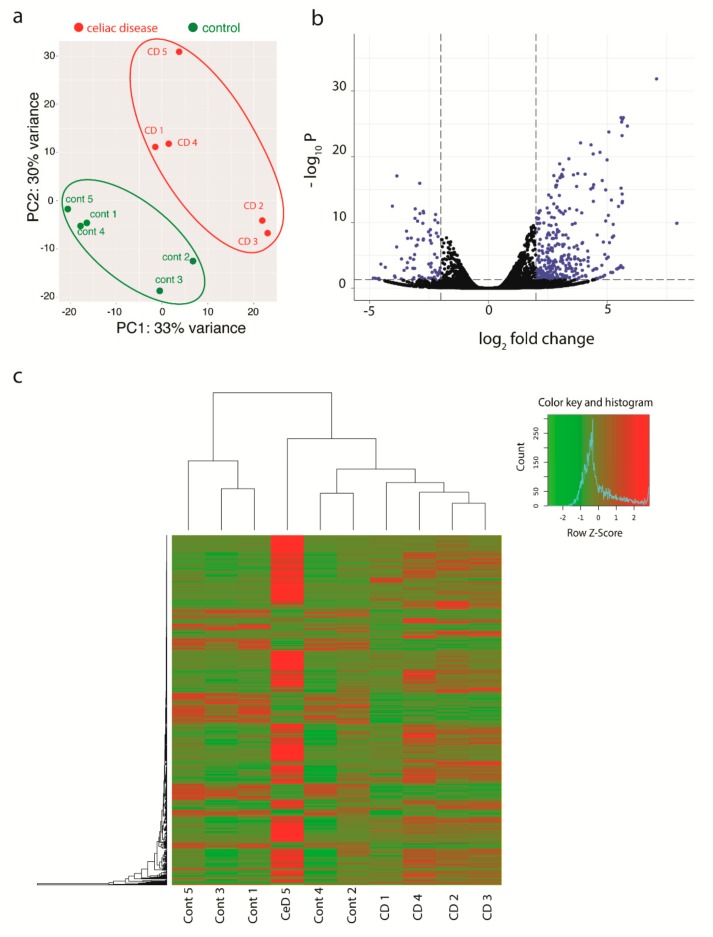
Multivariate visualization of the analyzed genes. (**a**) PCA scatter plot PCA plot showing variance between CD patients’ samples and non-celiac control samples (PC1), and heterogeneity between the five biological replicates in each group (PC2). (**b**) Volcano plot representing the results of the analysis. Each dot representing one gene, and the blue highlighted genes were significantly differentially expressed (*p*-value < 0.05) and log fold change cut-off 2. (**c**) Heatmap shows hierarchical clustering of genes on the left side and the clustering of the samples on the top. The histogram represents the expression data of the significant differentially expressed genes (green is the down-regulated and red is the up-regulated).

**Figure 2 cells-09-00516-f002:**
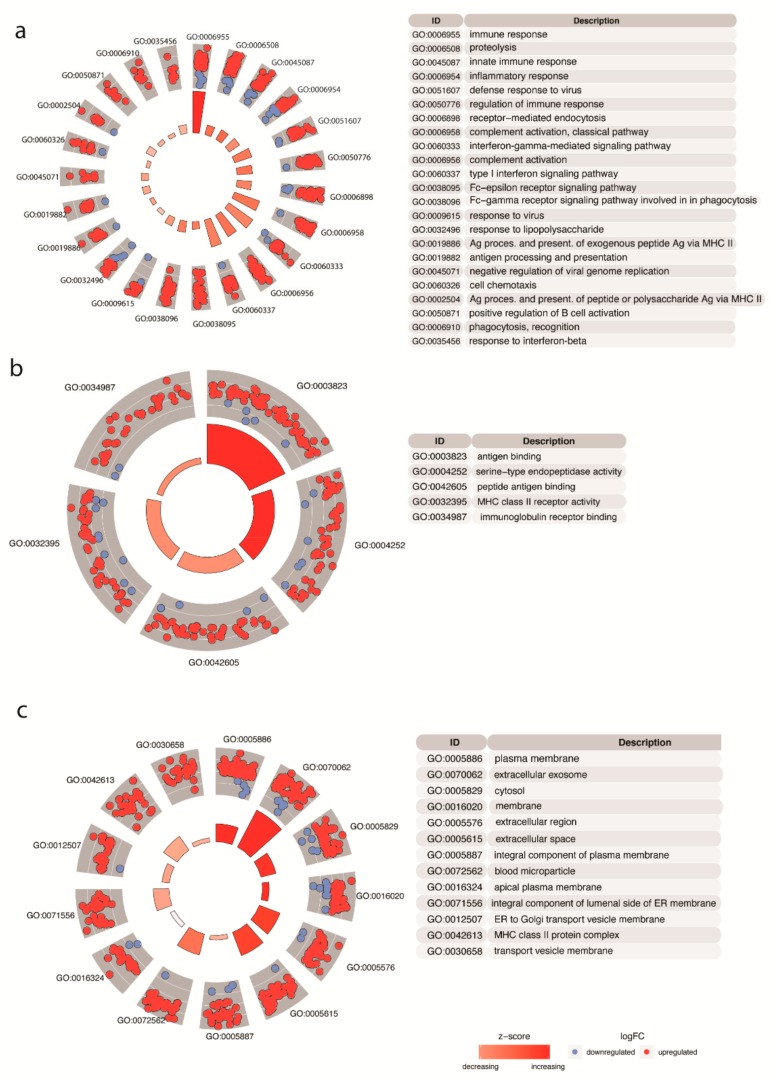
Results of functional enrichment analysis. The outer circle shows a scatter plot of each term, including their corresponding genes (blue dots down-regulated and red dot up-regulated). Z-score indicates the tendency to increase or decrease each of the gene ontology (GO) terms based on the ratio of the differentially expressed genes. Only the significant terms are displayed in the three main categories: (**a**) biological process, (**b**) cellular components, and (**c**) molecular function.

**Figure 3 cells-09-00516-f003:**
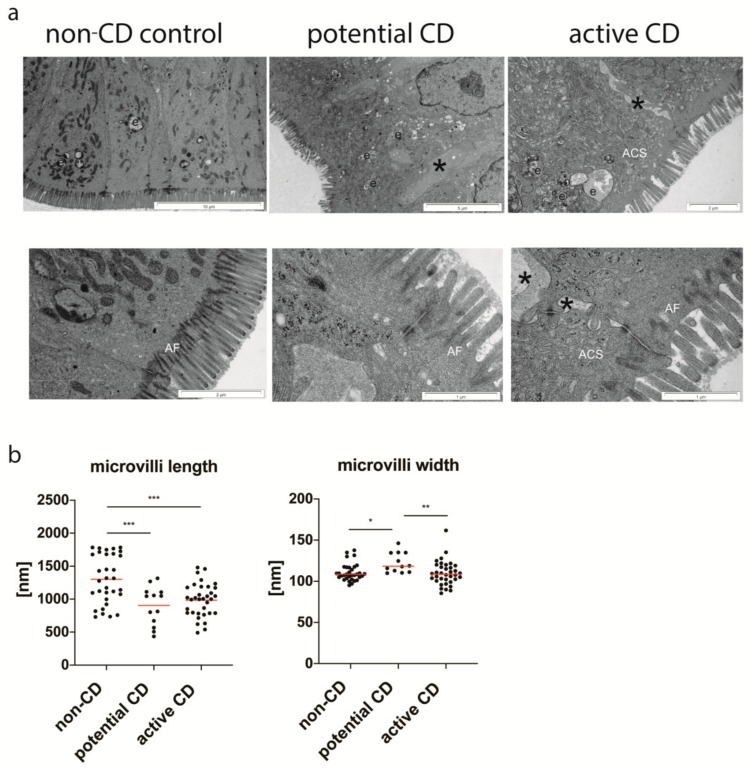
Ultrastructure of small intestine enterocytes from the non-CD control group, patients with potential and active CD (**a**). Cross-sections of non-CD enterocytes with anchoring filaments (AF), endosomes (e) and no intracellular dilatations between cells present in the control group. Enterocytes in potential and active CD exhibit numerous endosomes (e) and tubules of an apical canicular system (ACS), and dilated intercellular spaces (*). The length and width of the brush border microvilli (**b**). Measurements were done with the use of the morphometric iTEM program (Olympus) in 10 selected epithelial areas at a magnification of ×60,000, and at least 3 values/patient were obtained. All measurements are presented. Statistical analysis was performed with the use of one-way ANOVA with Tukey correction for multiple comparisons. * *p* < 0.05, ** *p* ≤ 0.01, *** *p* ≤ 0.001.

**Figure 4 cells-09-00516-f004:**
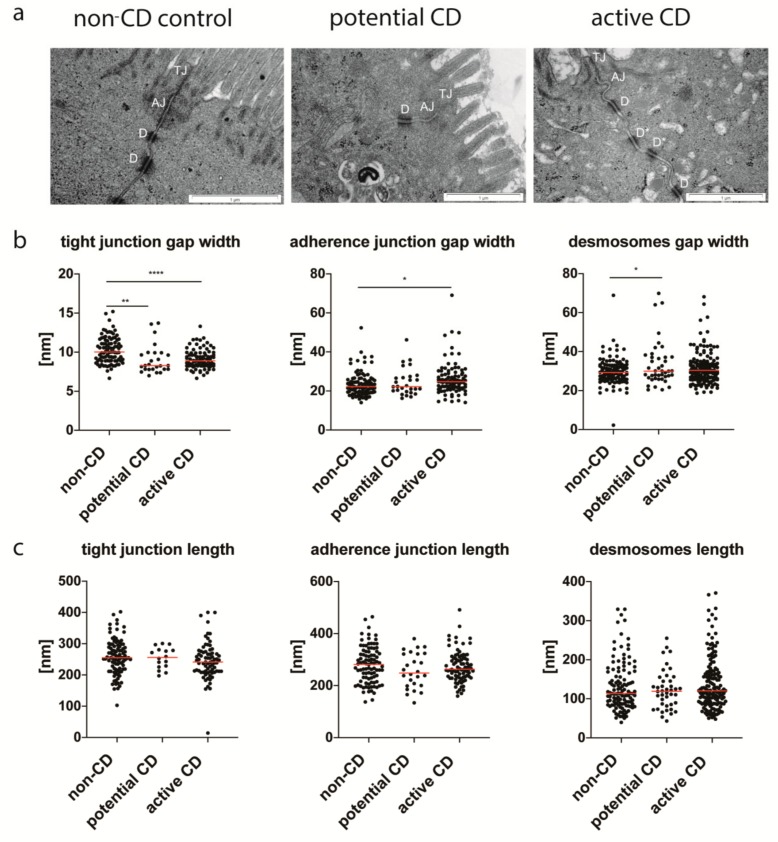
The proximal region of the enterocytes with tight junctions (TJ), adherence junctions (AJ) and desmosomes (D) from the non-CD control group, patients with potential and active CD (**a**), and ultrastructural features of intercellular junctions (**b**,**c**). Desmosomes with an incorrect asymmetrical structure (D*) present in a patient with active CD. The widths (**b**) and lengths (**c**) of EJC were measured using the morphometric iTEM program (Olympus) at a magnification of ×60,000. Measurements were done in 10 selected epithelial areas through longitudinally sectioned intercellular junctions, and at least 5 values of each type of junction/patient were obtained. All measurements are presented. Statistical analysis was performed with the use of one-way ANOVA with Tukey correction for multiple comparisons. * *p* < 0.05, ** *p* ≤ 0.01, **** *p* ≤ 0.0001.

**Table 1 cells-09-00516-t001:** Summary of the most relevant Sequence Read Archive (SRA) projects for celiac disease (CD).

No	Title	Project Accession Number	No of Experiment
1	Expression of long non-coding RNAs in autoimmunity and linkage to enhancer function and autoimmune disease risk genetic variants	PRJNA357628	15
2	Transcriptome of celiac disease	PRJNA327491	42
3	Chronic inflammation permanently reshapes tissue-resident immunity in celiac disease (human)	PRJNA509448	34
4	RNA sequencing of intestinal mucosa in celiac patients	PRJNA528755	42
5	The interplay between IL-15, gluten, and HLA-DQ8 drives the development of coeliac disease in mice (human)	PRJNA556711	95

**Table 2 cells-09-00516-t002:** Characteristics of CD patients enrolled in ultrastructural studies.

No of Patient	Sex	Age [in Years]	The Level of Anti-tTG2-IgA/[AU/mL]	Histology	Diagnosis
1.	Male	8	10	Marsh 0	Potential CD
2.	Male	7	19	Marsh 1	Potential CD
3.	Female	6	24	Marsh 1	Potential CD
4.	Female	10	31	Marsh 2	Active CD
5.	Male	7	54	Marsh 3a	Active CD
6.	Male	7	>100	Marsh 3b	Active CD
7.	Female	14	>100	Marsh 3b	Active CD
8.	Female	10	>100	Marsh 3b	Active CD
9.	Male	9	>100	Marsh 3b	Active CD
10.	Male	9	Negative*	Marsh 3b	Active CD
11.	Male	6	>100	Marsh 3b	Active CD
12.	Female	3	>100	Marsh 3c	Active CD

Histological changes were determined using the modified Marsh-Oberhuber classification. The active CD was recognized in accordance with the ESPGHAN 2012 guidelines. *Patient No. 10 had IgA deficit and positive anti-tTG2-IgG (>100 AU/mL). All patients with potential CD (patients No. 1, 2, 3) had positive anti-EMA-IgA antibodies in the following titers 1: 5, 1:50, 1: 5, respectively.

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
