# Peer review of "Transcriptional and Ultrastructural Analyses Suggest Novel Insights into Epithelial Barrier Impairment in Celiac Disease"

_cells, 2020, doi:10.3390/cells9020516_

Round 1
Reviewer 1 Report
Sowinska and colleagues investigated epithelial barrier in celiac disease by transcriptional and ultrastructural analysis. This manuscript has provided meaningful information, but there are some limitations, which await to be improved.
Authors thoroughly performed mRNA NGS analysis. However, besides suggested disturbance of intestinal barrier on the level of epithelial cells, which is known, authors did not dig out other information from the data. Authors could have found more clues. It is not clear that how authors performed the statistical analysis. 12 patients and 9 controls were included in this study, however, according to the dot plot, there are definitely more points. I don’t think it is fair to use multiple data points from each patient to preform statistical analysis. For Figure 4a, author should provide a magnified version of the figures.
Author Response
We would like to thank the reviewers for the revision of our manuscript that helped us to improve the manuscript. Below please find our point by point response to questions or comments. In the manuscript all changes are marked in red.
Question no 1
Authors thoroughly performed mRNA NGS analysis. However, besides suggested disturbance of intestinal barrier on the level of epithelial cells, which is known, authors did not dig out other information from the data. Authors could have found more clues.
Answer to question no 1:
Analysis of mRNA NGS data gives a lot of possibilities and information that might be used in several ways. We agree that information gained from mRNA NGS analysis presented in our manuscript could be expanded, and more clues could be found. However, for the purpose of our manuscript, we have focused our analysis on GO terms that describe molecular functions. Analysis of mRNA NGS data directed in this way allowed us to justify the second part of our manuscript, where we aim to show structural differences in epithelial cells from celiac patients and non-celiac controls.
Remaining information from mRNA NGS analysis of epithelial cells that have not been addressed in this manuscript will be used for the preparation of further manuscripts, where functional data will support mRNA NGS analysis.
Question no 2
It is not clear that how authors performed the statistical analysis.
Answer to question no 2
Information about the statistical analysis of mRNA NGS is updated in the manuscript. A new paragraph in section 2.5 from Material and Methods was added (lines 134-139):
" Generalized-linear model based on the negative binominal distribution to estimate the dispersion parameter across all genes with assumption of common dispersion for all genes was applied for RNA-seq data analysis. The default trimmed mean of M-values (TMM) method was used for normalization, and adjustment of multiple comparisons (with a false discovery cut-off <0.05) was performed using Benjamini-Hochberg correction”
Question no 3
12 patients and 9 controls were included in this study, however, according to the dot plot, there are definitely more points. I don't think it is fair to use multiple data points from each patient to preform statistical analysis.
Answer to question no 3:
It is correct that 12 patients and 9 controls were included in our second part of manuscript, where we aim to present structural changes and differences in epithelial cells from celiac patients and non-celiac controls. The multiple data points from each patient is a result of performed measurements for each patient. And now the description of done measurements was explained exactly and corrected in section 2.4 – Material and Methods (lines 122-125) as well as in the description below Figures 3 and 4.
Section 2.4:
"Ultrastructural analyses were performed in 10 selected tissue areas, and only epithelial structures that were precisely longitudinally sectioned were measured. At least 5 values of each type of intercellular junction/patient, and at least 3 values associated with microvilli/patient were obtained. The results were presented in nm."
Figure 3, line 215-2217:
"Measurements were done with the use of the morphometric iTEM program (Olympus) in 10 selected epithelial areas at a magnification of x 60 000, and at least 3 values/patient were obtained. All measurements are presented."
Figure 4. line 225-229:
"Measurements were done in 10 selected epithelial areas through longitudinally sectioned intercellular junctions, and at least 55 values of each type of junction/patient were obtained. All measurements are presented."
Question no 4
For Figure 4a, author should provide a magnified version of the figures.
Answer to question no 4:
Due to the editorial limits of the total file not exceeded 120 MB, it was not possible to provide higher image resolution at the figure 4a in the manuscript (one original image in "tiff format" is about 36MB). Microscopical magnification used in manuscript figures (x60 000) was optimal to visualize all structures of complex epithelial junctions (including TJ, AD and desmosomes) in the same picture. We are in contact with the Editor to send original images in tiff format to include them as a supplementary materials. Increased Figure 4 is included as pdf, however after conversion the file the quality of photos decreased.

Reviewer 2 Report
This study shows that an increased intestinal permeability, due to abnormality of epithelium, might play a role in CD onset, although the hypothesis that an increased passage of gluten peptides can precedes CD development is not supported by the results.
This is an interesting and solid work and I have no major methodological concerns.
I have only some doubt about the classification of pediatric subjects:
1. Marsh 2 (only one patient) should not be considered as an active CD in absence of villous atrophy.
2. The three cases of potential CD had tTG-IgA values too low (< 10 fold from normal value) to be classified as potential CD. To confirm this clinical suspect the dosage of anti-endomysium IgA had to be also performed.
Author Response
We would like to thank the reviewers for the revision of our manuscript that helped us to improve the manuscript. Below please find our point by point response to questions or comments. In the manuscript all changes are marked in red.
Question no 1
Marsh 2 (only one patient) should not be considered as an active CD in the absence of villous atrophy
Answer to question no 1:
The diagnosis of active CD was recognized in accordance with the current guidelines of the European Society for Pediatric Gastroenterology Hepatology and Nutrition from 2012 for pediatric patients. All experts agree that histological changes are not so specific marker for CD, and now the most important are specific autoantibodies (especially anty- tTG). In the situation when anty-tTG-IgA are positive, both Marsh 2 and Marsh 3 are recognized as the active CD. That is why in this study we decided to include this one patient with Marsh 2. Our analyses did not show differences when this patients was out of the study. However, we agree that this was not exactly explained in the manuscript, and now we describe more precisely the basis for diagnosis in section 2.3, line 99-100
2.3. Patients' biopsy samples
"The active CD was recognized in accordance with the current guidelines of European Society for Pediatric Gastroenterology Hepatology and Nutrition (ESPGHAN) from 2012, i.e. in patients with positive immunoglobulin (Ig) A antibodies against tTG2 (anti-tTG2-IgA) or in case of IgA deficit with positive anti-tTG2-IgG, and with histological changes described at least as Marsh 2 [19]."
Question no 2
The three cases of potential CD had tTG-IgA values too low (< 10 fold from normal value) to be classified as potential CD. To confirm this clinical suspect the dosage of anti-endomysium IgA had to be also performed.
Answer to question no 2:
Anti-EMA antibodies were determined in all patients, but following ESPGHAN guidelines for CD, we did not show these results in the manuscript. As a result of the suggestion, we added values of anti-EMA antibodies for each potential CD patient in the description of patients in Table 2 (line 128-131). We also described it in section 2.3 (line 105-107). In Discussion section we emphasized that both type of antibodies EMA and tTG-IgA were done in patients with potential CD (line 291-292).
Now it is:
Section 2.3
"In addition, antibodies against endomysium (EMA) with the use of indirect immunofluorescence assay (Euroimmun, Lubeck, Germany) were determined in all patients with potential CD. A titer of ≥ 1: 5 was considered as a positive result".
Table 2:
"All patients with potential CD (patients No. 1, 2, 3) had positive anti-EMA antibodies in the following titers 1: 5, 1:50, 1: 5, respectively".

Round 2
Reviewer 1 Report
No further comment.